# Application of Pelvic Circumferential Compression Devices in Pelvic Ring Fractures—Are Guidelines Followed in Daily Practice?

**DOI:** 10.3390/jcm10061297

**Published:** 2021-03-21

**Authors:** Valerie Kuner, Nicole van Veelen, Stephanie Studer, Bryan Van de Wall, Jürgen Fornaro, Michael Stickel, Matthias Knobe, Reto Babst, Frank J.P. Beeres, Björn-Christian Link

**Affiliations:** 1Department of Orthopaedic and Trauma Surgery, Cantonal Hospital Lucerne, 6000 Luzern, Switzerland; Nicole.vanVeelen@luks.ch (N.v.V.); Bryan.VandeWall@luks.ch (B.V.d.W.); Matthias.Knobe@luks.ch (M.K.); Frank.Beeres@luks.ch (F.J.P.B.); Bjoern-Christian.Link@luks.ch (B.-C.L.); 2Medical Faculty, University of Zurich, 8091 Zurich, Switzerland; StephanieStuder@gmx.ch; 3Department of Radiology, Cantonal Hospital Lucerne, 6000 Luzern, Switzerland; Juergen.Fornaro@luks.ch; 4Department of Emergency Care, Cantonal Hospital Lucerne, 6000 Luzern, Switzerland; Michael.Stickel@icloud.com; 5Department of Health Science and Medicine, University of Lucerne, 6002 Luzern, Switzerland; Reto.Babst@luks.ch

**Keywords:** pelvic ring fracture, PCCD, position, associated injuries

## Abstract

Early administration of a pelvic circumferential compression device (PCCD) is recommended for suspected pelvic trauma. This study was conducted to evaluate the prevalence of PCCD in patients with pelvic fractures assigned to the resuscitation room (RR) of a Level I trauma center. Furthermore, correct application of the PCCD as well as associated injuries with potential clinical sequelae were assessed. All patients with pelvic fractures assigned to the RR of a level one trauma center between 2016 and 2017 were evaluated retrospectively. Presence and position of the PCCD on the initial trauma scan were assessed and rated. Associated injuries with potential adverse effects on clinical outcome were analysed. Seventy-seven patients were included, of which 26 (34%) had a PCCD in place. Eighteen (23%) patients had an unstable fracture pattern of whom ten (56%) had received a PCCD. The PCCD was correctly placed in four (15%) cases, acceptable in 12 (46%) and incorrectly in ten (39%). Of all patients with pelvic fractures (*n* = 77, 100%) treated in the RR, only one third (*n* = 26, 34%) had a PCCD. In addition, 39% of PCCDs were positioned incorrectly. Of the patients with unstable pelvic fractures (*n* = 18, 100%), more than half either did not receive any PCCD (*n* = 8, 44%) or had one which was inadequately positioned (*n* = 2, 11 %). These results underline that preclinical and clinical education programs on PCCD indication and application should be critically reassessed.

## 1. Introduction

About 20% of polytrauma patients have a pelvic injury [1], with an estimated incidence of about 23 per 100,000 persons per year [2,3]. The examination of pelvic stability is part of the primary survey of trauma patients as an unstable pelvic ring fracture may result in severe intra- or retroperitoneal bleeding [4,5,6]. If, based on the mechanism of injury or clinical findings, an unstable pelvic ring injury is suspected, current guidelines recommend applying a pelvic circumferential compression device (PCCD) [4,7,8] to minimize the risk of intrapelvic haemorrhage and promote coagulation by realigning the pelvic ring and therefore reducing the pelvic volume [6,9,10,11,12]. A further option to reduce anterior diastasis is simple internal rotation of the lower extremities, which can be held by tape as reported by Gardner et al. However, this technique is problematic if the lower limbs are unstable due to long bone fractures [13,14,15]. Ideally, the PCCD is applied in the preclinical setting directly at the site of the accident. Typically, it is left in place until either the injury is ruled out or treatment is initiated [16,17]. The PCCD should be positioned over the greater trochanters to allow for optimal transmission of forces via the proximal femur to the pelvis to reduce anterior diastasis [18,19]. Potential disadvantages of PCCDs such as skin necrosis and nerve lesions have been described in case reports [20,21,22,23,24,25,26].

This study aimed to evaluate the prevalence and quality of PCCD application in pelvic fractures of patients assigned to the resuscitation room (RR) in a level I trauma center in Switzerland as well as to assess potential adverse effects in relation to associated injuries using a PCCD.

The hypothesis of this study is that the majority of patients with pelvic fractures treated in the resuscitation room at this level I trauma center have a correctly positioned PCCD in place and few adverse effects occur.

## 2. Materials and Methods

This article was written in accordance with the STROBE statement [27]. The study was approved by the Ethics Committee of Northwest- and Central Switzerland (project ID 2018-00411). The need for informed consent was waived.

### 2.1. Study Design, Setting and Participants

The imaging and electronic patient records of all consecutive patients treated in the RR of a level one trauma center in Switzerland were evaluated retrospectively for the years 2016 and 2017. Resuscitation room management at this trauma center follows a defined algorithm, which is based on the Advanced Trauma Life Support (ATLS) algorithm [7,28,29] and has been adjusted according to the Whitebook Medical Care of the Severely Injured of the German Society for Trauma Surgery [30]. If hemodynamically stable, all patients receive a whole-body computer tomography (CT) after the primary survey has been completed. All patients with a traumatic pelvic fracture diagnosed in the whole-body CT were included in this study. Fragility and subacute fractures were excluded. Fractures obtained from low-energy trauma such as a fall from standing height were defined as fragility fractures [31]. In fragility fractures the ligament structures remain intact, so there is no major bleeding and therefore no role for PCCDs. Subacute fractures were defined as fractures with visible callus formation with or without previous documentation of the fracture.

### 2.2. Data Measurement and Variables

Demographic data of each patient were collected from the electronic medical records (Medfolio, Nexus AG, Donaueschingen, Germany) as well as the Swiss Trauma Register (STR) and the register of the German Society for Trauma Surgery (DGU). In these registers there are six possible trauma mechanisms to allocate the case to: car, motorcycle, cyclist, pedestrian, fall from height or other (such as explosion or blow). Any fall that was from higher than standing height was defined as a fall from height. The New Injury Severity Score (NISS) total was compiled and divided into groups <16 and ≥16 points on the NISS scale [32,33,34]. To be able to classify the fractures and evaluate the positioning of the PCCD patients had to have had a CT. Hemodynamically unstable patients, who required immediate intervention prior to CT imaging were therefore excluded.

Images were viewed using the Picture Archiving and Communication System (Phönix- PACS GmbH, Freiburg i.Br, Germany). All fractures were classified according to the modified Tile AO classification by an orthopaedic resident and revised by a fellowship-trained pelvic surgeon [35,36]. Unstable fractures were defined as Tile B1, B3, C1, C2 or C3. Stable fractures were defined as Tile A1, A2, A3 and B2. Type B2 fractures (ipsilateral internal rotation injury) were classified as stable fractures, as these fractures are caused by an internal rotation force and the volume of the pelvis is not enlarged by the injury. Further, it is assumed that the anatomy of these fractures is restored by the elastic recoil of the pelvis [35]. For these reasons, such injuries do not benefit from the use of a PCCD. Fractures that could not be classified according to Tile, such as acetabular fractures classified according to Judet and Letournel [37] and sacral fractures according to Denis [38] I-III plus sacral transversal, sacral U- and H-shaped fractures, were rated as stable fracture patterns, since these do not benefit from the use of a PCCD. Further, it was assessed whether the fracture involved the neuroforamina.

In the catchment area of this trauma center two different types of PCCD, the T-PODTM (Cybertech Medical, Laverne, CA, USA) and the SAM Pelvic-sling IITM (SAM Medical Products, Tualatin, OR, USA) are in use [39,40].

Presence upon arrival in the emergency room (yes or no), type (the T-POD lap loop or the SAM Pelvic-sling II) and position of a PCCD were assessed on the CT scans. The position of the PCCD was rated as ‘correct’ if it covered both greater trochanters completely, ‘acceptable’ if the PCCD partly covered the greater trochanters and ´incorrect´ if the PCCD did not cover the greater trochanters at all (Figure 1) [18,19].

All associated injuries based on the CT findings like presence of bladder injury, neuroforaminal fracture involvement, vascular injury, pelvic hematoma and based on medical records of presence of neurogenic bladder disorder, posttraumatic peripheral neurologic injury of the lower extremities and skin necrosis were recorded.

Bladder injury, neurogenic bladder disorder, neuroforaminal fracture involvement with simultaneous posttraumatic peripheral neurologic injury of the lower extremities and vascular injury were defined as associated injuries which could be aggravated by the application of a PCCD.

### 2.3. Statistical Methods

The data collected were analysed using SPSS (IBM^®^ SPSS^®^ Statistics 24, IBM, Armonk, NY, USA). Mean values, medians, standard deviation and percentages were calculated. The Fisher exact test was used for the statistical analysis of associated injuries and PCCD presence (significance level *p* < 0.05). Subgroup analysis was tempted stratified for fracture stability by using the Fisher exact test. The Chi- Square Test was used to analyze the relationship between the positioning of PCCD and the associated injuries (significance level *p* < 0.05).

## 3. Results

A total of 730 patients were admitted to the RR during the study period. Eighty-two (11%) had a pelvic fracture. All patients were hemodynamically stable enough to receive a CT prior to any intervention. Four patients with a fragility fracture and one patient with a subacute pubic fracture were excluded leaving a total of 77 patients (Figure 2). The demographic data are summarized in Table 1. All patients sustained their injuries by blunt trauma.

Twenty-six (34%) patients had a PCCD in place at the time of the CT examination (Figure 2). Out of these, 24 PCCDs had been placed preclinically, for the remaining two no reliable documentation on the time of application was found. Eighteen (69%) patients had the T-POD device and eight (31%) the SAM pelvic sling II in place.

Fracture classification according to Tile is listed in Table 2. There were no C1 fractures in the cohort. Fifty-eight (75%) patients could be classified according to Tile. The remaining 25% had acetabular or sacral fractures which could not be classified according to Tile and were therefore assigned to the stable fracture patterns (Table 2).

Eighteen (23%) patients had an unstable fracture pattern, of which ten (56%) received a PCCD, while 16 (27%) of the patients with a stable fracture had one applied. This leaves a total of 51 patients without PCCD, 43 (73%) of the patients with stable fractures, and eight (44%) of those with unstable fractures.

In total ten (13%) patients correctly received a PCCD based on the fracture pattern, eight (10%) should have received a PCCD, 43 (56%) did not get a PCCD since there was no indication for application and 16 (21%) received a PCCD although indication was not given (Figure 2).

The position of the PCCD was correct in four (15%) cases, acceptable in 12 (46%) and incorrect in ten (39%) (Table 3). Regarding the ten patients with unstable fractures, PCCD position was correct in one patient, acceptable in seven patients and inadequate in two (Table 3).

The type of PCCD and the respective positioning are listed in (Table 4).

The associated injuries are listed in Table 5. One patient had two associated injuries (neurogenic bladder disorder and neuroforaminal fracture involvement with simultaneous posttraumatic peripheral neurologic injury of the lower extremities). In this collective no PCCD related adverse effects such as skin necrosis were registered.

Patients with applied PCCD showed a significantly higher rate of associated injuries like bladder injury (*n* = 3), neurogenic bladder disorder (*n* = 3), neuroforaminal fracture involvement with simultaneous posttraumatic peripheral neurologic injury of the lower extremities (*n* = 1), traumatic vascular injury (*n* = 7) than patients without PCCD according to the Fisher exact test (*p* = 0.0075) (Table 6).

Within the group of PCCD patients, the subgroup analysis stratified for fracture stability showed no significant difference regarding the incidence of associated injuries for unstable fracture patterns (Fisher exact test *p* = 0.3137) (Table 6), while there was a significant association in the stable fracture group (Fisher exact test *p*= 0.0278) (Table 6).

There was a statistically significant relationship between the positioning of PCCD and the occurrence of associated injuries according to the Chi- Square Test (X2(1, N = 77) = 5.0667, *p* < 0.05) (Table 7).

## 4. Discussion

In this retrospective study comprising of 77 patients with a pelvic fracture only one third (26/77; 34%) had a PCCD applied. The position of the PCCD was correct in four (15%) cases, acceptable in 12 (46%) and incorrect in ten (39%).

Several recent studies are in accordance with these findings that only a minority of patients with pelvic ring fractures are preclinically treated with a PCCD [41,42,43,44]. The largest cohort so far is described by another Swiss study overlooking a period of 6 years that found a PCCD applied in only 552/2366 (23%) of the cases [45]. Also in a recent study of Vaidya et al. one third of patients with unstable pelvic fractures did not receive a PCCD [41]. In this study, the rate was even higher with 44% (8/18).

According to studies that mainly examined external rotation injuries and the reduction of symphyseal diastasis by PCCD, the PCCD needs to be positioned over the greater trochanters for optimal efficiency [18,19,46]. Retrospective studies which analysed the position of the PCCD in relation to the trochanters have shown that the position is incorrect in up to 50% [18,47,48,49,50].

In this cohort, 39% had incorrect positioning. Williamson et al. found a similar sub- optimal placing of PCCD in 43.5%, 39.7% were placed superior and 3.8% inferior to the greater trochanter line [48]. Other studies demonstrate a PCCD misalignment of up to 50% [49,50].

Of the 10 patients with an unstable fracture and a PCCD seven (70%) received the T-POD and three (30%) the SAM pelvic sling II. No superiority of one PCCD model over the other could be found by Knops et al. [51]. This study cohort was too small to evaluate the superiority of one type of device over the other.

The retrospective observational design and the relatively small size of the study population limit the conclusive strength of this study. The small number of patients who received a PCCD does not allow for statistical evaluation regarding the type of PCCD applied and limited the statistical power of subanalysis regarding a connection between PCCD positioning and potential adverse effects on associated injuries. The correction for other confounders was not possible due to the low sample size, therefore, it was chosen to only stratify for stable and unstable fractures. A low rate of accurate indication and correct PCCD application has been reported by several authors [18,41,42,43,44,45,47,48,49,50].

The evaluation of pelvic ring stability at the site of an accident is often hampered by several factors such as patient consciousness, environmental circumstances and clothing. These inherent factors seem to limit the accuracy of the clinical evaluation of pelvic stability. Studies assessing these difficulties concluded that since clinical stability testing of the pelvis showed low sensitivity [46,47,48], the accident mechanism was a more relevant factor influencing the decision on whether or not a PCCD is indicated [52,53,54]. However, the trauma mechanism might be unclear in a substantial proportion of cases emphasising the challenges encountered in the field. The classification into stable fractures, for which the PCCD is not beneficial, and unstable fractures, for which the indication is given, was used for retrospective analysis. This classification was based on the review of CT images of the fractures, which is naturally impossible for preclinical staff who must rely on clinical signs and the mechanism of injury to judge the stability of a pelvic injury. It does however, highlight the fact that unstable pelvic fractures were undersupplied with PCCDs in this study population and that there is room for improvement.

The low rate of correctly positioned PCCD could be addressed by sensitising and instructing paramedics and RR personnel on accurate identification of anatomical landmarks. Williamson et al. defined the correct position of the PCCD as a position between the tip of the greater trochanter and the inferior border of the lesser trochanter. He found a significantly higher risk for misplacement of the PCCD if the distance between those anatomic landmarks was small (<8.9 cm) and in females [48]. Due to smaller body size in females the palpable bony mass of the greater trochanter is also smaller. This might cause additional problems in positioning the PCCD correctly. Familiarization with different PCCD models through training seems to be an additional factor [44,55], as different types of PCCDs will require knowledge for their correct positioning in relation to palpable landmarks. Obesity or secondary dislocation during patient transport of the PCCD are also factors that can hamper correct positioning.

The accident kinematics as well as the preclinical assessment and initial clinical examination should trigger the suspicion of a pelvic ring injury [4,7]. If an unstable pelvic ring injury is suspected, stabilization using a PCCD is an effective temporary measure in an emergency situation [4,7,26]. Additional advantages of a PCCD are pain control, haemorrhage control [39,56], reduced transfusion requirement [12,57], reduction in the length of hospitalization [12,57] and decreased mortality [12]. PCCDs are non-invasive and can be applied rapidly on the scene of an accident [40].

In the preclinical phase, the PCCD is the gold standard for pelvic stabilization. The stabilization of pelvic fractures with severe and persistent hemodynamic instability can be achieved by invasive procedures in the RR such as the C-clamp [58,59,60] for pelvic ring lesions of type C and by external fixation [61,62,63,64] for the B-type [65,66]. PCCDs show an equivalence to invasive procedures like the C-clamp, which requires more user knowledge, time, training and equipment [67]. It also provides comparable stability to invasive procedures such as the external fixators [68]. However, in contrast to the PCCD, the c-clamp and external fixator can be used for definitive treatment.

In respect to the fracture pattern, to our knowledge there is no evidence in the literature that compression of internal rotation injuries or acetabular fractures with a PCCD may cause adverse effects [10,11,42,51]. However, known potential disadvantages of PCCD application are pressure decubitus or aggravating nerve compression with long lasting application, as described in case reports and studies that analyze pressure measurement on models or healthy subjects [22,69,70,71,72]. No skin necrosis complication occurred in this patient population. Due to short transport distances and prompt patient care in the catchment area of central Switzerland the paramedic should not hesitate to apply a PCCD because of potential risk of skin necrosis. However, the current study found more potential aggravation of associated injuries with PCCD application (Table 6). Despite these differences being statistically significant, the total number of patients included in this study population was relatively low, prohibiting the ability to draw a sound conclusion. It however demonstates the need for further investigation into this topic.

Interestingly, this potential aggravation of associated injuries was statistically significant for stable fractures (Table 6), but not for unstable fractures (Table 6) in the subgroup analysis. The definition of a stable (A1-3, B2) and unstable (B1, B3, C1-3) fracture chosen for this study may be a potential confounder in this sub-analysis. It was assumed that the anatomy in B2 fractures is restored by the elastic recoil of the pelvis near to normal so it does not benefit from the use of a PCCD. The compressive effect of a PCCD, however, may reproduce or even aggravate the initial accident mechanism, therefore potentially leading to further injuries. Out of the five patients with stable fractures who received a PCCD and had associated injuries two were B2 fractures. Both of these cases had a vascular injury. The other three were two A2 and one A3 fracture. Further studies are needed to address risk factors in terms of PCCD effects. These could give rise to balance potential life- saving benefits against potential adverse events [26].

Conversely, the rate of PCCD in patients with pelvic fractures in general and specifically in patients with unstable pelvic ring fractures is rather low. Naturally, there seems to be room for improvement in education. In Central Switzerland, paramedics receive lessons on how to correctly apply a PCCD as part of their training, this includes an instructional video and a handout. Qualified paramedics must complete a total of 40 h of mandatory annual training which includes the application of PCCDs. In the latest (10th) edition of the ATLS Student Manual, for the first time a video on PCCD application including anatomic landmark instructions is linked to the App [7]. These educational efforts underline the observed knowledge gap regarding the indication for and correct application of PCCDs. Perhaps more practical training with instruction of additional palpable anatomic landmarks besides the level of the trochanters such as the relation of the upper belt border to the anterior superior iliac spine is needed to avoid grossly incorrect PCCD positioning and could increase the rate of correct PCCD applications.

## 5. Conclusions

Only one third (34%) of patients with pelvic fractures assigned to the RR had a PCCD placed. Moreover, of these, 39% were applied incorrectly. These results, in accordance with similar results from the recent literature, clearly demonstrate the need for focused preclinical and clinical education programs on when and how to apply a PCCD. 

The observed rate of potential aggravated adverse effects of PCCD’s seems to be higher in the treated group irrespective of the fracture pattern. Higher patient numbers are needed to balance the live saving benefits of PCCD’s against potential adverse effects of its application. 

In the meantime, the PCCD remains the gold standard, however by rising the awareness of correct indication and positioning in the catchment area of this level I trauma center potential adverse effects could likely be minimised and benefits increased.

## Figures and Tables

**Figure 1 jcm-10-01297-f001:**
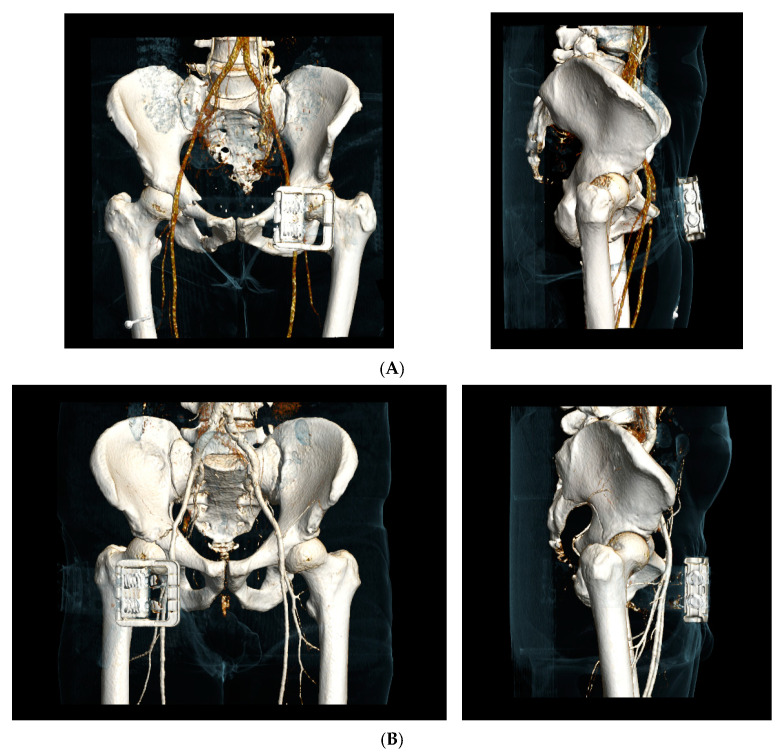
Pelvic circumferential compression device (PCCD) positioning (example of the SAM Pelvic-sling II). (**A**) Correct positioning of a PCCD at the level of the trochanters; (**B**) Acceptable positioning of a PCCD with partial coverage of the trochanters; (**C**) Incorrect positioning of a PCCD without any covering of the trochanters.

**Figure 2 jcm-10-01297-f002:**
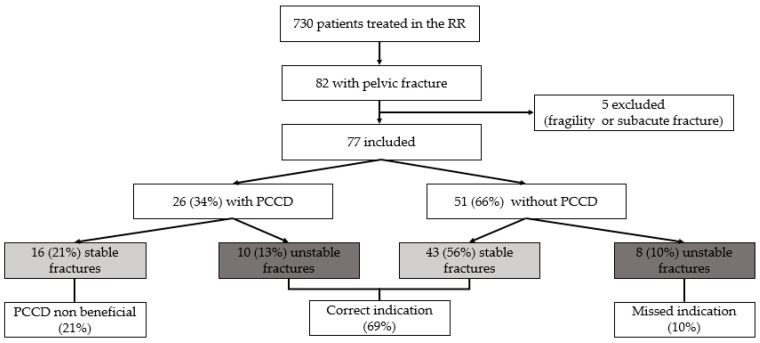
Included patients, prevalence of a PCCD and indication for PCCD.

**Table 1 jcm-10-01297-t001:** Demographic data.

Demographic Data	N	Mean
Sex		
Male	49 (64%)	
Female	28 (36%)	
Age		50 years (range 14–93, SD ± 21.2)
Trauma mechanism		
Blunt	77 (100%)	
Type of accident		
Car	6 (8%)	
Motorcycle	13 (17%)	
Cyclist	6 (8%)	
Pedestrian	8 (10%)	
Fall from height	36 (47%)	
Other (like blow, explosion)	8 (10%)	
NISS		20 (range 4–66, SD ± 16)
NISS≥/<16		
≥16	54 (70%)	
<16	23 (30%)	
Intensive medical treatment		
Yes	40 (52%)	
No	37 (48%)	
Survivors	71 (92%)	
Death	6 (8%)	
Length of hospital stay (days)		11 (0–98 days, SD ± 14)

**Table 2 jcm-10-01297-t002:** Prevalence of fractures and pelvic circumferential compression device (PCCD).

Fracture Type	Total *n* = 77	PCCD Placed *n* = 26	PCCD Not Placed *n* = 51
Pelvic ring fractures according to Tile stable and unstable	58 (75%)		
A1	2 (3%)	0	2
A2	16 (21%)	4	12
A3	2 (3%)	1	1
B1	5 (6%)	2 ^1^	3 ^2^
B2	20 (26%)	5	15
B3	8 (10%)	5 ^1^	3 ^2^
C1	0 (0%)	0	0
C2	2 (3%)	2 ^1^	0
C3	3 (4%)	1 ^1^	2 ^2^
acetabular fracture	17 (22%)	6	11
isolated sacral fracture	2 (3%)	0	2
additional femoral neck fracture	3 (in 1 bilateral)	1	2
additional Pipkin fracture	2	2	0

^1^ Ten (13%) correct indications for PCCD. ^2^ Eight (10%) unstable fractures, indication for PCCD.

**Table 3 jcm-10-01297-t003:** PCCD positioning.

Position of PCCD	Unstable Factures *n* = 10	Stable Fractures *n* = 16
correct	1 (10%)	3 (19%)
acceptable	7 (70%)	5 (31%)
incorrect	2 (20%)	8 (50%)

**Table 4 jcm-10-01297-t004:** PCCD type (T-POD or SAM Pelvic Sling) in unstable fractures.

Position of T-POD (*n* = 7)		Position of SAM Pelvic Sling (*n* = 3)	
correct	1 (14%)	correct	0
acceptable	5 (71%)	acceptable	2 (67%)
incorrect	1 (14%)	incorrect	1 (33%)

**Table 5 jcm-10-01297-t005:** Associated injuries of pelvic fracture.

Associated Injury	*n* = 77	Fracture Type
Bladder injury	3 (4%)	B1
		B2 + Denis I + acetabular anterior wall
		acetabular anterior column, posterior hemitransverse
Neurogenic bladder disorder	3 (4%)	B2 + sacrum fracture H- type
		C3 + sacrum fracture H- type + acetabular anterior column ^1^
		acetabular anterior column
Neuroforaminal fracture involvement	14 (18%)	A3 + transverse sacrum fracture
		B2 + Denis II
		B2 + Denis II
		B2 + Denis II
		B2 + Denis II
		B2 + Denis II
		B2 + Denis III
		B2 + sacrum fracture H- type
		B3 + transverse sacrum fracture
		B3 + sacrum fracture U- type
		C2 + Denis II
		C3 + sacrum fracture H- type
		Denis II + acetabular anterior colum with hemitransverse
		Denis II + acetabular 2 colum fracture
Nerve lesion of the lower extremities	4 (5%)	C3 + sacrum fracture H- type
		sacrum fracture H- type
		acetabular transverse fracture
		acetabular 2 colum fracture
Neuroforaminal fracture involvement + posttraumatic peripheral neurologic injury of the lower extremities	1 (1%)	C3 + sacrum fracture H- type + acetabular anterior column ^1^
Traumatic vascular injury	7 (9%)	A3 + transverse sacrum fracture
		B1
		B1
		B2 + Denis II
		B2 + Denis III
		C2 + Denis II
		acetabular transverse fracture
Pelvic hematoma	19 (25%)	A2
		A2
		A2 + Denis I
		A3 + transverse sacrum fracture
		B1 + Denis I
		B2 + Denis I
		B2 + Denis I
		B2 + Denis I
		B2 + Denis II
		B2 + sacrum fracture H- type
		B2 + acetabular anterior wall
		B3
		B3 + Denis I
		B3 + Denis I
		C2 + Denis I
		C3 + sacrum fracture H- type + acetabular anterior colum
		acetabular anterior colum with hemitransverse
		acetabular 2 colum fracture
		left acetabular transverse + posterior wall, right acetabular posterior wall

^1^ One patient showed two associated injuries.

**Table 6 jcm-10-01297-t006:** Contingency table of all documented associated injuries in relation to applied PCCD. (**a**) Contingency table of all documented associated injuries in patients with unstable fractures in relation to applied PCCD. (**b**) Contingency table of all documented associated injuries in patients with stable fractures in relation to applied PCCD.

	**PCCD**	**No PCCD**	
Associated injury	9 (69 %)	4 (31%)	13
No associated injury	17 (27 %)	47 (73%)	64
	26	51	77
(**a**)			
	**PCCD**	**No PCCD**	
Associated injury	4 (80%)	1 (20%)	5
No associated injury	6 (46%)	7 (54%)	13
	10	8	18
(**b**)			
	**PCCD**	**No PCCD**	
Associated injury	5 (62.5%)	3 (37.5%)	8
No associated injury	11 (22%)	40 (78%)	51
	16	43	59

The Fisher exact test statistic value is 0.0075. The result is significant at *p* < 0.05. (**a**) The Fisher exact test statistic value is 0.3137. The result is not significant at *p* < 0.05. (**b**) The Fisher exact test statistic value is 0.0278. The result is significant at *p* < 0.05.

**Table 7 jcm-10-01297-t007:** Contingency table of all documented associated injuries in relation to the position of the applied PCCD.

	PCCD Correct or PCCD Acceptable	PCCD Incorrect or No PCCD	
Associated injury	6 (43%)	8 (57%)	14
No associated injury	10 (16%)	53 (84%)	63
	16	61	77

The chi-square statistic is 5.0667. The *p*-value is 0.02439. Significant at *p* < 0.05.

## Data Availability

The data presented in this study are available on request from the corresponding author. The data are not publicly available due to privacy reasons.

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
