# Peer review of "Application of Pelvic Circumferential Compression Devices in Pelvic Ring Fractures—Are Guidelines Followed in Daily Practice?"

_jcm, 2021, doi:10.3390/jcm10061297_

Round 1

Reviewer 1 Report

Many thanks to the authors for addressing this important topic. The paper retrospectively examines the frequency of external emergency pelvic stabilisation (PCCD). On the one hand, the pPCCD device is now widely used, on the other hand, its use is partly inflationary. The authors thus address a relevant topic. 

Abstracts:
Well written, relevant data presented.

Introduction: 

no comments

Material and methods:

Appropriate methods, presentation of PCCD position very good even for the reader not familiar with it.

Results:

  • Some minor remarks:
  • Were patients included who were haemodynamically unstable on arrival?
    - If so, how many and in how many was emergency pelvic intervention (angiography, packing) used?
    - In some cases a subsequent correction of the PCCD is described, was a PCCD also subsequently applied in the shock room in patients with unstable pelvic injuries?

Line 190: Typo: Space between table and 7

discussion:

adequate discussion
 236: typo et al.
 Limitation correctly and well presented.
Further comments: 
- Current studies consider the PCCD to be equivalent to the previously used investment methods ( Audretsch CK, Mader D, Bahrs C, Trulson A, Höch A, Herath SC, Küper MA; Working Group on Pelvic Fractures of the German Trauma Society. Comparison of pelvic C-clamp and pelvic binder for emergency stabilization and bleeding control in type-C pelvic ring fractures. Sci Rep. 2021 Jan 27;11(1):2338. doi: 10.1038/s41598-021-8174; 
Zeckey C, Cavalcanti Kußmaul A, Suero EM, Kammerlander C, Greiner A, Woiczinski M, Braun C, Flatz W, Boecker W, Becker CA. The T-pod is as stable as supraacetabular fixation using 1 or 2 Schanz screws in partially unstable pelvic fractures: a biomechanical study. Eur J Med Res. 2020 Jul 18;25.

I think a paragraph on the biomechanical advantages and the equivalence to invasive procedures should be highlighted in the discussion, also for readers who are not familiar with this procedure.

Conclusion:

No comments 

Summa summarum, this is a solid paper that will be of interest to every medical professional working in ambulance services and the emergency department, as well as to trauma specialists. 

Reviewer 2 Report

I thank the editor and the authors to review this very interesting manuscript regarding the Application of Pelvic Circumferential Compression Device in Pelvic Ring Fractures. The authors present a retrospective analysis of the prevalence of compression device in pelvic ring fractures. This kind of study fits into the scope of the Journal of Clinical Medicine. However, I have some reservations and recommend a rejection.

  • Ll.30-32: misleading conclusion: “Remarkably more than half of patients with unstable pelvic fractures did not receive a PCCD or had it positioned inadequately”. à “more than half of the patients with pelvic fractures did not receive a PCCD and more than one third had it positioned inadequately.”
  • Ll.38: literature: „more than 25% of polytraumatized patients have a pelvic ring fracture” Notfallmanagement instabiler Beckenverletzungen. Notfall &318 Rettungsmedizin 7:151–160. https://doi.org/10.1007/s10049-004-0646-x
  • Date of pubblication 2004 and 2011, please search for more current literature
  • Ll.46-48: please look for literature regarding the rotation of the femora
  • Ll.50-51: “Recently, doubts about the benefit of routine usage of PCCDs in the trauma setting have arisen” wrong assumptionà your literature consists of case reports including 8 patients and only 2 current case reports: 3cases from 2020, 1 case from 2017, 1 case from 2009, 4 cases from 2005, 1 case from 2003
  • L. 55: please add a hypothesis
  • Ll.62-63: a retrospective study over a longer time period would bring up more substantial data
  • L. 69: “Fragility and subacute fractures were excluded”. It is not evident how these kinds of fractures were defined by CT. Was the trauma mechanism an inclusion criteria? FFP III and IV are also unstable pelvic ring fractures Dtsch Arztebl Int 2018; 115: 70-80. DOI: 10.3238/arztebl.2018.0070
  • Ll. 95-97: please cite literature underlining the correct position of the circumferential compression device
  • Table 1: please give a more precise definition of the trauma mechanism: high energy trauma according to the ATLS guidelines? What was the definition of the fall from the height? 3m?
  • Table 2 an l. 139: for the prehospital stuff it is not possible to classify pelvic ring fractures according to Tile or according to Judet and Letournel. Even after a CT scan there might be different classification results in a clinical setting of a trauma center. A prehospital correct indication for application of a PCCD is the clinical suspicion for a pelvic ring fracture after a high energy trauma.

  • Fig. 2: the percentages do not match à left arm:
  • Unnecessary application: in the prehospital setting it might be difficult - without having a mobile CT-scan – to define a pelvic ring fracture as stable
  • Fig. 2: how is it possible to get to know whether the pelvic fracture is stable before the CT-scan?
  •  
  • Ll.187-190: what is the exact perception of a significant Chi- Square Test without a statement about the strength and direction? What was the aim of this statistical test?
  • L. 209: “In this cohort, 39% had incorrect positioning.” à 10% missed indications in unstable fractures, 21% “unnecessary application”
  • Ll.217-222: “small size … limited the statistical power of subanalysis”: please increase the study size for more substantial statistical results, otherwise this study is a retrospective descriptive study only. Are there any clinical consequences of the subgroup analyses? A subgroup analysis without a possible consequence or a basis for discussion remains a nice statistical tool. Please do not fish for statistical significant results.
  • 246-252: “If an unstable pelvic ring injury is suspected, stabilization using a PCCD is an effective temporary measure in an emergency situation”. Why do you discuss in the results an “unnecessary application”?
  • Ll.263-274: The study size is very small. It is difficult to draw statistical conclusion for the general population.
  • Ll. 195-196 + 289-290: “Only one third had a PCCD”. What is the point? Only one third of the patients get a PCCD, so the amount is too little. On the other side you discuss that 16 patients of these 26 patients had an “unnecessary application of PCCD”.
  • Please clarify the results of table 6b and discuss them. Is PCCD associated with an aggravation of associated injuries? What is about the possible bias that the prehospital stuff put on a PCCD after getting information about the trauma mechanism?
  • Ll.292-294: “The observed rate of potential aggravated adverse effects of PCCD’s seems to be higher in the treated group irrespective of the fracture pattern.” Is this a significant result? What is the conclusion for the prehospital stuff and medical stuff handling with pelvic ring fractures? When should we put on a PCCD? What are the recommendation and the knowledge gain of this study compared to the studies you discussed?: Zingg T, Piaget-Rossel R, Steppacher J, et al (2020) Prehospital use of pelvic circumferential compression devices in a physician-based emergency medical service: A 6-year retrospective cohort study. Sci Rep 10:. https://doi.org/10.1038/s41598-020-62027-6; Prasarn ML, Small J, Conrad B, et al (2013) Does Application Position of the T-POD Affect Stability of Pelvic Fractures? Journal of Orthopaedic Trauma 27:262–266. https://doi.org/10.1097/BOT.0b013e31826913d6; Naseem H, Nesbitt P, Sprott D, Clayson A (2017) An assessment of pelvic binder placement at a UK major trauma centre. Annals 100:101–105. https://doi.org/10.1308/rcsann.2017.0159;
  •  

Round 2

Reviewer 2 Report

I thank the editor and the authors to review the revised version of the manuscript “Application of Pelvic Circumferential Compression Device in Pelvic Ring Fractures”. After substantial changes, the manuscript has improved greatly. All comments were checked and answered.

Current literature was selected (point 2) and the femoral rotation was listed (point 3). Due to the changes regarding points 1 and 4 the message of the manuscript changed substantially. Adding a hypothesis and defining the trauma mechanism and thus the inclusion criteria were necessary improvements. The revision of the discussion -including ll.259-265 showing that patients with unstable pelvic ring fractures are undersupplied- effects the manuscript positively. Rephrasing the conclusion and avoiding judging expressions improved the manuscript. The figures are now better to understand. Thank you for adding the statement in ll. 340-342. Further investigations are necessary, however the statistical analysis remains weak.

All in all, after substantial revisions the present manuscript may be published in JCM.